# Effect of Nitric Oxide on Browning of Stem Tip Explants of *Malus sieversii*

Chen Yang, Jiangfei Liu, Xin Qin, Yangbo Liu, Mengyi Sui, Yawen Zhang, Yanli Hu, Yunfei Mao * and Xiang Shen *

College of Horticulture Science and Engineering, Shandong Agricultural University, Tai'an 271000, China; yangchensdau@163.com (C.Y.); liujiangfeisdau@163.com (J.L.); qinxinsdau@163.com (X.Q.); liuboyangsdau@163.com (Y.L.); smy09042000@163.com (M.S.); zyw18905363662@163.com (Y.Z.); ylhu8612@163.com (Y.H.)

* Correspondence: myfmyf@sdau.edu.cn (Y.M.); shenx@sdau.edu.cn (X.S.)

**Abstract:** Browning is a major problem in the tissue culture of woody plants. Previous studies have shown that nitric oxide (NO) plays a role in regulating plant responses to stress, but its effect on browning in the tissue culture of Malus remains unclear. This study aimed to investigate the impact of exogenous NO donor sodium nitroprusside (SNP) on the browning of *Malus sieversii* stem tip explants. The results demonstrated that the addition of 50 μM SNP significantly reduced explant browning. Further analysis revealed that exogenous NO decreased the browning index (BD) and levels of malondialdehyde (MDA), hydrogen peroxide ($H_2O_2$), and superoxide anion radical $O_2^-$. Additionally, NO treatment increased the activity of antioxidant enzymes such as superoxide dismutase (SOD), catalase (CAT), and ascorbate peroxidase (APX). NO treatment also enhanced the activity of phenylalanine ammonia lyase (PAL), which contributed to the accumulation of flavonoids and inhibited the activity of polyphenol oxidase (PPO) and peroxidase (POD), which are key enzymes involved in the browning process. Furthermore, 50 μM SNP significantly promoted the accumulation of non-enzymatic antioxidants such as proline (Pro), soluble sugar (SS), and soluble protein (SP). Therefore, the results suggest that NO is able to counteract excessive reactive oxygen species (ROS) damage by enhancing both the enzymatic and non-enzymatic antioxidant systems, resulting in a reduction in browning in stem tip explants. Consequently, an improvement in the in vitro propagation efficiency of *Malus sieversii* shoot tip explants can be achieved.

**Keywords:** sodium nitroprusside (SNP); *Malus sieversii*; tissue culture; reactive oxygen species (ROS); antioxidants; browning; browning control

## 1. Introduction

*Malus sieversii* (Ledeb.) M.Roem, a wild apple tree, has a wide distribution in Central Asian countries, including Kazakhstan, Kyrgyzstan, Tajikistan, Uzbekistan, Turkmenistan, and the Tianshan Mountains in western Xinjiang of the People's Republic of China [1]. Extensive research has shown that *Malus sieversii* serves as the ancestor of modern cultivated apple (Malus domestica) [2,3]. Over thousands of years of evolution, *M. sieversii* has developed numerous desirable characteristics and exhibits strong resistance against biotic and abiotic stresses [4]. Therefore, it is commonly used as a rootstock to enhance disease resistance and environmental adaptability in cultivated apples, as well as to cultivate new varieties. *M. sieversii* is considered an important source of disease resistance, stress resistance, and new fruit traits.

Currently, the population of *M. sieversii* is greatly declining due to human activities, the invasion of alien species, and climate change. Additionally, the risk of genetic integrity loss is increasing with the expansion of cultivated apple planting and the adoption of cross-pollination reproduction [5]. Consequently, there is an urgent need to develop effective

measures for the protection of *M. sieversii*'s genetic resources in order to prevent the further decline of its germplasm.

Plant micropropagation has emerged as a widely utilized technique for the large-scale propagation and preservation of endangered plant germplasm resources. Direct bud induction, considered an efficient micropropagation method, allows for the production of healthy clones that maintain the same characteristics as the female parent [6,7]. The most optimal regeneration system for the leaf and stem tip of *M. sieversii* has been previously reported [8]. However, the issue of browning poses a significant challenge in the micropropagation of *M. sieversii* using stem tip explants. Numerous studies have demonstrated that woody plants, with a high phenol content, are more susceptible to browning, which can inhibit callus induction, adventitious bud regeneration, and ultimately result in explant death under severe circumstances [9,10]. Consequently, the problem of browning must be addressed when employing stem tip explants in tissue culture.

Callus browning during phenolic oxidation is attributed to the collective action of phenolic compounds, reactive oxygen species (ROS), and various oxidases [11]. The cutting process during explant preparation, particularly in the culture stage, is a crucial factor contributing to explant browning and increased ROS levels. Research indicates that the production rates of superoxide anion radical ($O_2^-$) and hydrogen peroxide ($H_2O_2$), along with other ROS, significantly elevate during plant tissue culture. Excessive ROS levels generally exert a detrimental impact on explant growth [12]. Moreover, oxidases such as SOD, POD, and PPO also play substantial roles in callus formation and are closely associated with browning [13,14].

Nitric oxide (NO) plays diverse regulatory roles in plant growth and development. It promotes the lateral bud germination of explants, induces callus formation, and delays callus stress, among other functions [15,16]. Notably, NO exhibits antioxidant activity within plant cells. It safeguards against oxidative stress by directly reacting with ROS, enhancing the activity of antioxidant enzymes, and reducing ROS accumulation. Exogenous NO has been extensively reported to regulate plant responses to various biotic and abiotic stresses by mitigating oxidative stress [17]. Sodium nitroprusside (SNP), an exogenous NO donor, is frequently employed in tissue culture due to its growth-promoting properties [18].

Currently, there are no reports on the impact of sodium nitroprusside on the browning of *M. sieversii* stem tip explants during tissue culture. In this study, we aimed to explore the inhibitory effect of exogenous NO on browning in tissue culture using *M. sieversii* stem tip explants. Considering the unique characteristics of *M. sieversii* and the physiological role of NO, we investigated the effect of NO on browning inhibition in *M. sieversii* explants by assessing various physiological and biochemical parameters throughout the tissue culture process. Conducting this research holds great significance for the efficient propagation and commercial production of resistant rootstocks.

## 2. Results

*2.1. Effects of NO Treatment on Shoot Tip Browning Phenotype and Browning Index during Tissue Culture*

As depicted in Figure 1a, the browning index of the stem tip in both the SNP treatment group and the control group exhibited an increasing trend with the prolongation of tissue culture time. Notably, the browning index of the control group was significantly higher than that of the SNP treatment group, reaching 77% after 96 h of culture. This severe browning had a detrimental effect on the growth of the material, resulting in browning and death of the majority of the explants. Contrarily, in the SNP treatment group, the browning buds displayed only a small amount of yellow substance at the cut during the initial days after inoculation. The buds were primarily green, leading to a very low browning index within the first 12 h, all less than 20%. However, after 48 h, the browning at the wound site of the explant intensified, impeding growth and causing a rapid increase in the browning index. By 96 h, the browning index of the 10, 50, and 100 µM SNP treatment groups reached 47.32%, 21.01%, and 44.43%, respectively. The degree of browning is an indication of the

accumulation of oxidation products in plant materials during tissue culture. As observed in Figure 1b, the degree of browning deepened gradually with the extension of tissue culture time, aligning with the trend reflected by the browning index. This implies that treatment with 50 µM SNP effectively alleviates the progression of browning.

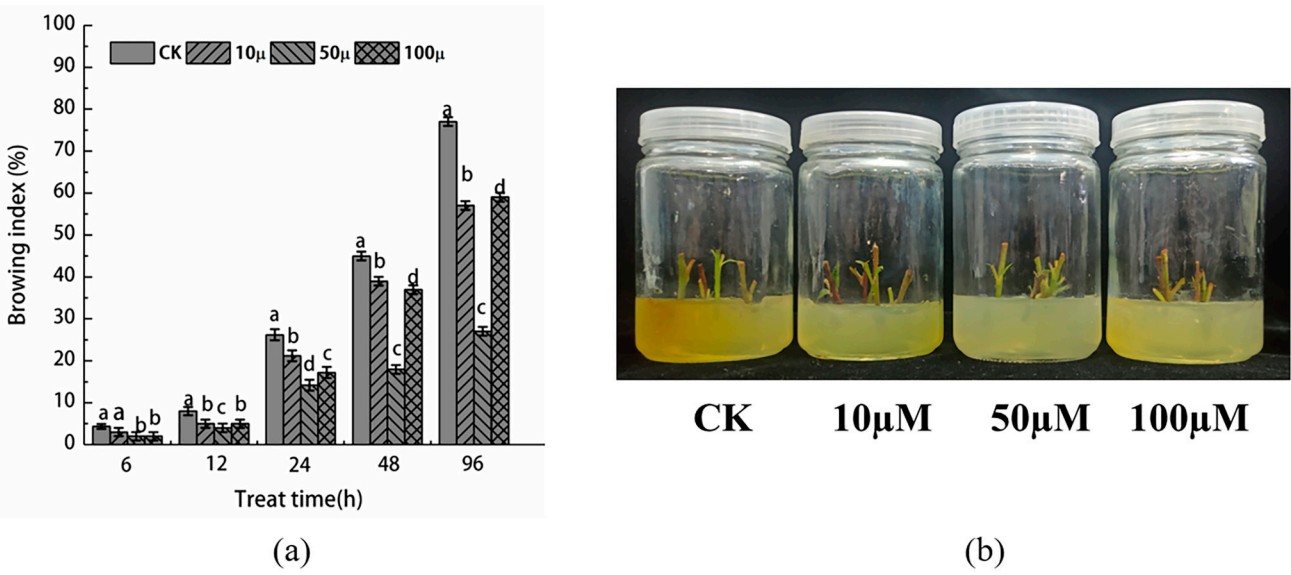

(a)                                                                                              (b)

**Figure 1.** (**a**) Browning index of stem tip explants of *Malus sieversii* treated with different concentrations of SNP and (**b**) browning phenotype after tissue culture for 96 h. According to Duncan's multiple range test, different letters indicate significant differences (*p* < 0.05).

### 2.2. Effect of NO Treatment on Enzymatic Antioxidant System of Explants

As illustrated in Figure 2a, the activities of SOD, POD, CAT, and APX in both the SNP treatment group and the control group initially increased and then decreased as the tissue culture time progressed. This pattern suggests that the excessive accumulation of reactive oxygen species resulting from browning contributed to damage to the active oxygen scavenging system in the stem tip of *Malus sieversii*. Furthermore, the activities of SOD, POD, CAT, and APX in the stem tip of the SNP treatment group consistently surpassed those of the control group. After 24 h of tissue culture, the activities of SOD and APX in the SNP treatment group's stem tip reached their peak, with the activities of 10, 50, and 100 µM SNP being 0.59, 0.82, and 0.49 times higher, respectively, than those of the control group's SOD.

The activity of POD exhibited a different pattern compared to SOD (Figure 2b). Within the first 24 h, the activity of POD in the SNP treatment group was higher than that of the CK group at 24 h but lower than that of the CK group. At 96 h, the activity of POD in the SNP treatment group was 0.19, 0.64, and 0.15 times lower than that of the CK group, respectively. Similar to SOD, the activity of CAT and APX followed a similar trend (Figure 2c,d). The activity of CAT reached its peak at 48 h, while the activity of APX peaked at 12 h. Compared to the CK group, the activity of CAT and APX in the 50 µM SNP treatment group increased by 66.4% and 105%, respectively. These findings indicate that different concentrations of SNP treatment effectively enhance the activity of the antioxidant enzyme system, thereby reducing damage caused by reactive oxygen species. Notably, the treatment with 50 µM SNP yielded the most favorable results.

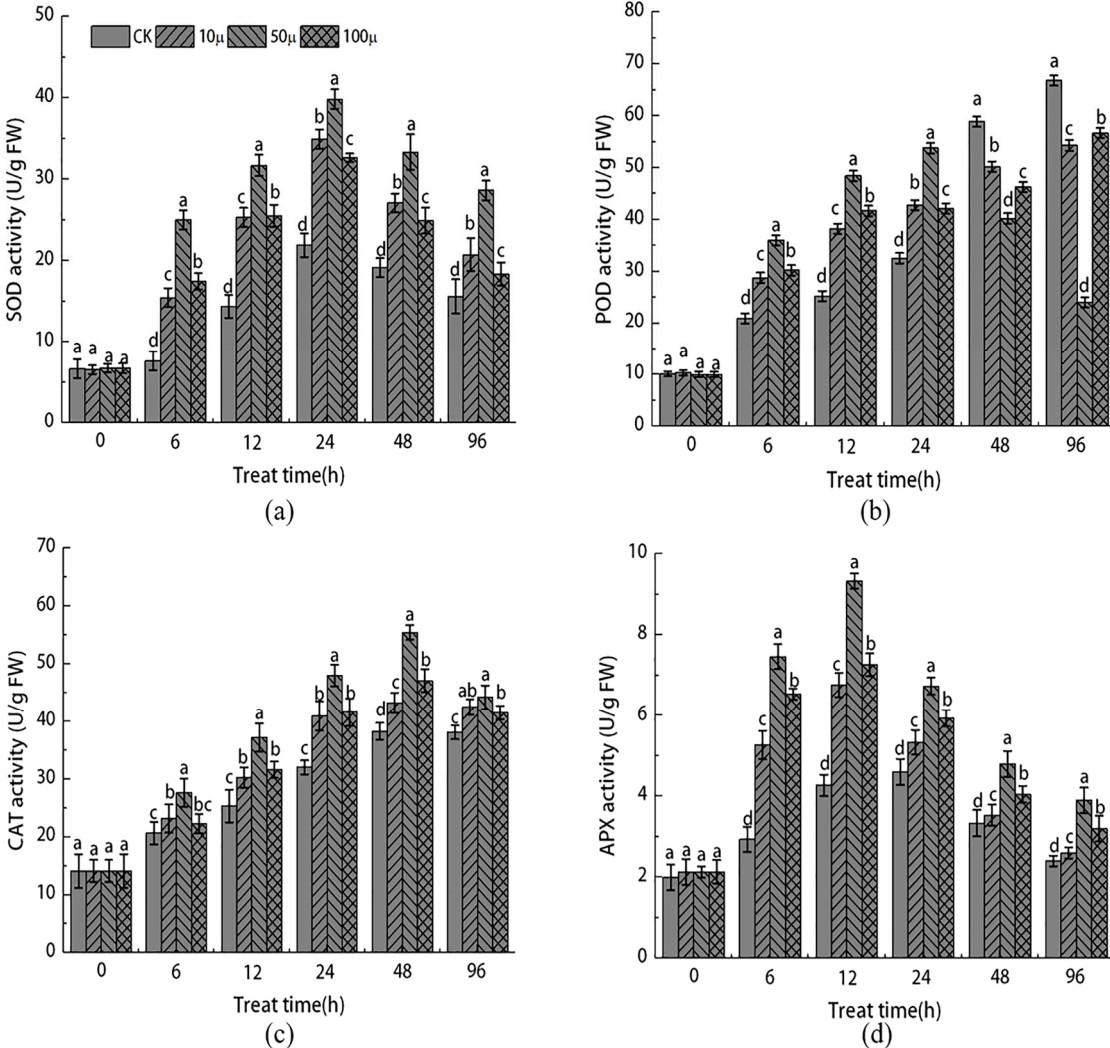

**Figure 2.** Effects of different concentrations of SNP on the activities of SOD (**a**), POD (**b**), CAT (**c**), and APX (**d**) in stem tip of *Malus sieversii*. The error bar represents the standard error of the average (*n* = 3). According to Duncan's multiple range test, different letters indicate significant difference (*p* < 0.05).

### 2.3. Effects of NO Treatment on Non-Enzymatic Antioxidants in Explants

Figure 3 demonstrates that as the tissue culture time progressed, both the CK and SNP-treated explants exhibited a similar pattern in the expression of non-enzymatic antioxidants, with a gradual increase that consistently surpassed the CK group. In terms of specific results, after treatment with 10, 50, and 100 μM SNP, the content of free proline (Pro) in the 50 μM SNP treatment group significantly increased by 17.4% compared to CK after 96 h of tissue culture (Figure 3a). In contrast, the 10 μM and 100 μM SNP treatment groups only experienced increases of 11.7% and 7%, respectively, compared to CK on the same medium.

This finding indicates that the 50 μM SNP treatment is most effective in increasing the content of Pro. Similarly, soluble protein (SP) (Figure 3b) and soluble sugar (SS) (Figure 3c) exhibited a similar pattern within 96 h of tissue culture, reaching their maximum levels at 48 h and 24 h, respectively. The results demonstrate that SNP treatment significantly enhanced the content of SS and SP in the explants, with the 50 μM SNP treatment having the most pronounced effect. Specifically, the 50 μM SNP treatment increased the content of SS and SP by 31.4% and 135.60%, respectively, compared to the CK group. These results suggest that SNP treatment can effectively respond to oxygen stress in explants by promoting the synthesis of Pro, SS, and SP. This helps in maintaining intracellular osmotic

pressure balance and eliminating the excessive production of oxygen free radicals caused by injuries.

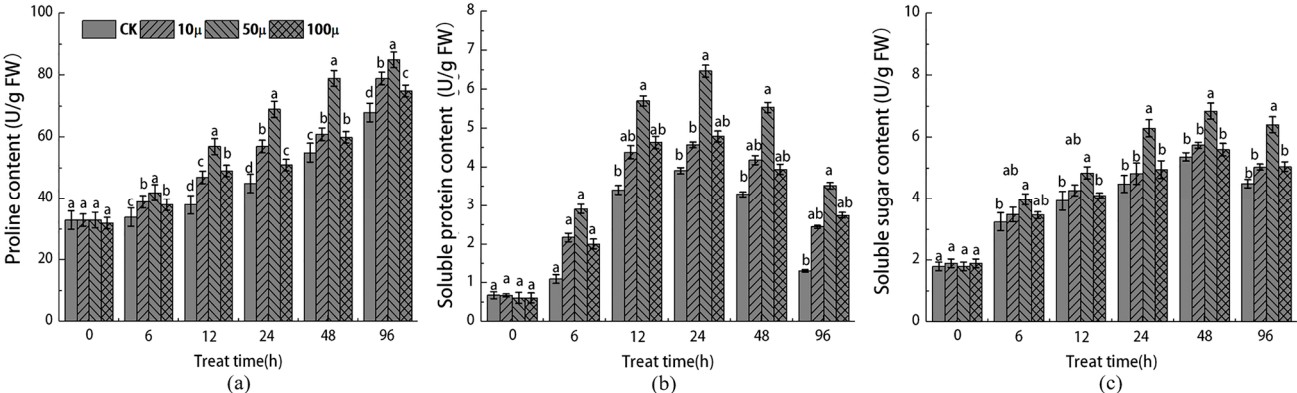

**Figure 3.** Effects of different concentrations of SNP on the contents of Pro (**a**), SP (**b**), and SS (**c**) in stem tip of *Malus sieversii*. The error bar represents the standard error of the average (*n* = 3). According to Duncan's multiple range test, different letters indicate significant difference (*p* < 0.05).

### 2.4. Effect of SNP on Markers of Oxidative Stress in Explants

The measurement of the MDA content is commonly used as an important indicator of cell membrane lipid peroxidation. In Figure 4a, we assessed the effect of different concentrations of SNP on the level of membrane lipid peroxidation in *Malus sieversii* stem tip cells. As the tissue culture time progressed, the MDA content in the stem tips of *Malus sieversii* increased in each comparison group. This suggests that the oxygen stress triggered by browning promoted membrane lipid peroxidation in the stem tips and caused damage to the membrane system. Compared to the control group, the content of MDA in the 10 μM, 50 μM, and 100 μM SNP treatment groups was lower, with a significant decrease compared to CK at 96 h. Furthermore, $H_2O_2$ and $O_2^-$ are commonly recognized as markers of oxidative damage in plant cells (Figure 4b,c). Throughout the tissue culture process, the concentration of $H_2O_2$ and $O_2^-$ gradually increased. However, in the 50 μM SNP treatment group, there was a significant decrease in the concentrations of both $H_2O_2$ and $O_2^-$. These results indicate that an appropriate concentration of SNP can inhibit the production of malondialdehyde and $H_2O_2$ in stem tip explants of *Malus sieversii*, with the best inhibitory effect observed at a concentration of 50 μM SNP.

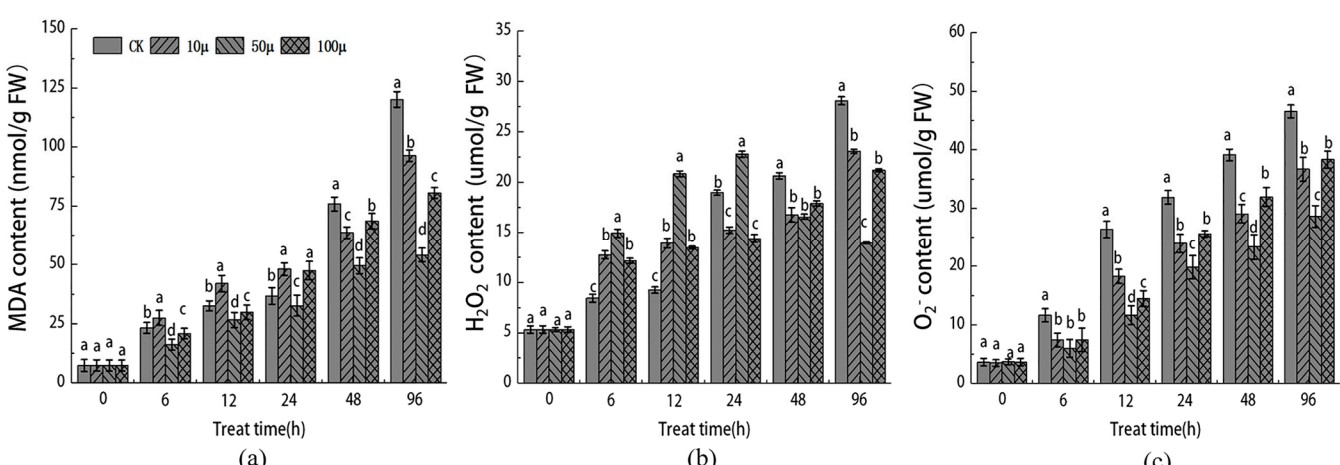

**Figure 4.** Effects of different concentrations of SNP on the contents of MDA (**a**), $H_2O_2$ (**b**), and $O_2^-$ (**c**) in stem tip of *Malus sieversii*. The error bar represents the standard error of the average (*n* = 3). According to Duncan's multiple range test, different letters indicate significant difference (*p* < 0.05).

### 2.5. Effects of SNP Treatment on the Activities of PAL and PPO and the Contents of Total Phenols and Flavonoids in Stem Tip Explants of Malus sieversii

As the tissue culture time extended, the PAL activity in the stem tip explants increased rapidly in each SNP treatment group, reaching its peak within the first 24 h. This peak was higher than that observed in the CK group. Subsequently, the PAL activity gradually decreased below the level of the CK group. In contrast, the PAL activity continued to increase in the CK group.

At a tissue culture duration of 96 h, the treatment with 50 μM SNP exhibited the most significant inhibition of PAL activity (Figure 5a). Conversely, the activity of PPO showed a different pattern compared to PAL. Throughout the entire tissue culture browning process, the activity of the SNP treatment group remained consistently lower than that of the CK group, displaying an upward trend. At the 96 h mark, the 50 μM SNP treatment led to a significant decrease of 58.5% compared to the CK group (Figure 5b).

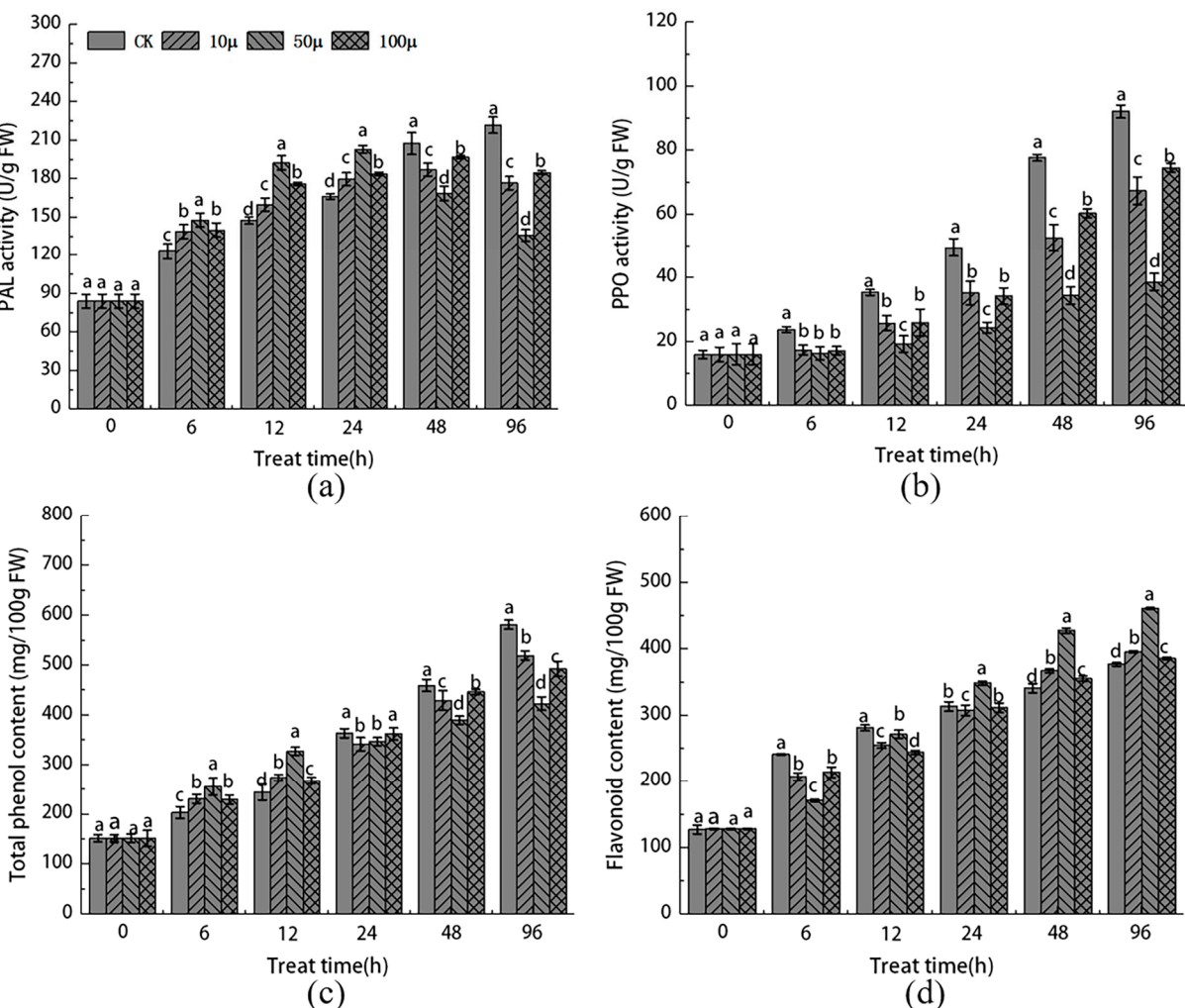

**Figure 5.** Effects of different concentrations of SNP on the activities of PAL (**a**) and PPO (**b**) and the contents of total phenols (**c**) and flavonoids (**d**) in stem tips of *Malus sieversii*. The error bar represents the standard error of the average value (*n* = 3). According to Duncan's multiple range test, different letters indicate significant differences (*p* < 0.05).

The content of total phenols in each SNP treatment group and the control group exhibited a gradual increase over time. After 24 h, the total phenol content in the SNP treatment group remained consistently lower than that in the control group. However, within the first 24 h, the SNP treatment group significantly promoted the increase in total phenols and flavonoids. Overall, the total phenol content in the stem tip explants of each

SNP treatment group decreased significantly in the early stage of tissue culture. By the 96 h mark, the total phenol content in the 50 μM SNP treatment group was significantly lower than that in the control group (Figure 5c). In contrast to the total phenol content, the SNP treatment notably enhanced the biosynthesis of flavonoids, particularly in the 50 μM SNP group within the first 24 h. The level of flavonoids was 0.26 times higher than that in the control group (Figure 5d). These findings demonstrate that SNP treatment effectively promotes phenylpropane metabolism in the stem tip of *Malus sieversii* in a short period of time. Additionally, it enhances the activity of the PAL enzyme and the accumulation of total phenols and flavonoids, while significantly inhibiting the key enzyme PPO involved in browning.

## 3. Discussion

This research examined the influence of exogenous NO on the browning of *Malus sieversii* stem tip explants in tissue culture and analyzed various indicators associated with this browning response. Given that woody plants, such as *Malus sieversii*, are rich in phenolic compounds, they are particularly susceptible to browning during tissue culture. Oxidation forms the core of the browning process, so minimizing the levels of reactive oxygen species and free radicals is crucial for preventing browning in explants [19,20]. NO, a crucial gaseous signaling molecule, is instrumental in plant growth and defense, helping plants resist both biotic and abiotic stressors [21]. SNP is frequently utilized as a NO donor. This study deployed three different SNP concentrations to assess their effect on browning in *Malus sieversii* shoot tip explants during tissue culture. The results indicated that SNP presented an inhibitory effect on browning compared to the control. Prior research indicates that varying concentrations of SNP exert differential promotive effects on the callus of *Dioscorea opposita*, with 40 μM of SNP optimally inhibiting browning. In contrast, a concentration of 150 μM SNP proved detrimental to the callus [22]. Moreover, SNP concentrations above 50 μM exacerbated browning in *Malus hupehensis* root callus [23], while the addition of 15 μM SNP to the culture medium was effective in reducing *Valeriana Jatamansi Jones* callus browning [24]. These findings suggest that an appropriate level of SNP can effectively neutralize oxygen free radicals formed during tissue damage, thereby mitigating browning. Our study further revealed that the beneficial impact of NO on oxidative stress is concentration-dependent. High levels of NO diminished its ameliorative effects, whereas lower concentrations were positively correlated with reduced oxidative stress. This indicates that NO's modulatory role as a signaling molecule may exhibit a dose-dependent response.

The concentration of MDA typically mirrors the extent of cellular membrane damage and oxidation inflicted by ROS. In this experiment, it was observed that the MDA levels in stem tip explants of *Malus sieversii* from the control group increased significantly in comparison to those treated with SNP, suggesting that lipid peroxidation, indicative of oxidative stress, led to enhanced membrane deterioration and browning. Conversely, when compared to the control (CK), treatment with NO markedly diminished the MDA content, demonstrating NO's capacity to effectively mitigate oxidative damage in explants. However, both low and high concentrations of NO failed to offer effective protection. The ability of SNP to reduce MDA under various stresses has been corroborated by studies on other plant species, including *cotton*, *wheat*, and *rice* [25–27]. Furthermore, SNP effectively curtailed the rise in the $O_2^-$ content and significantly inhibited $H_2O_2$ levels, thus demonstrably reducing $H_2O_2$ production. NO's antioxidant role includes ROS scavenging and lipid peroxidation reduction during oxidative stress [28]. Our research concurs with this, highlighting NO's capacity to eliminate oxygen free radicals and prevent further oxidative damage. This effect is likely because NO enhances the ROS-scavenging efficacy of the *Malus sieversii* antioxidant defense system.

Plants possess two antioxidant systems: enzymatic and non-enzymatic. In the enzymatic system, SOD, POD, CAT, and APX play key roles in RO5 detoxification under stress [29,30]. SOD can quickly disproportionate $O_2^-$ radical to $H_2O_2$ and oxygen, and

CAT, POD, and APX can catalyze the decomposition of $H_2O_2$ and balance the accumulation of ROS [31]. SNP treatment notably boosted the activity of these enzymes, indicating that NO's role in reducing both the browning index and browning intensity is tied to enhanced antioxidant enzyme activities. We also recorded increases in low-molecular-weight carbohydrates, like Pro, SP, and SS in *Malus sieversii* stem tip explants. These compounds are crucial osmotic regulators and are recognized as significant non-enzymatic antioxidants in plants. Proline accumulation often signifies plant resistance against stress, and SNP treatment at appropriate concentrations has been shown to foster proline accumulation, thereby reducing the callus browning of *Fi-cus religiosa* [32]. This study echoes these findings, showing that increased levels of Pro, SP, and SS aid in maintaining osmotic balance within cells. Additionally, our findings support the hypothesis that NO, as a vital signaling molecule, may regulate osmotic potential by modulating the nitrogen metabolism pathway under stress conditions [33].

NO plays a significant role in the regulation of phenolic metabolism and is involved in the callus development of explants. PAL is a key and rate-limiting enzyme in phenylpropane metabolism and plays a pivotal role in plant phenolic metabolism. Previous studies have demonstrated that NO treatment significantly enhances PAL activity in *sweetpotato* root callus [34], which is consistent with the findings of this study. PPO is an essential defense enzyme that plays a crucial role in various plant antioxidant stress responses [35]. When PPO in plant tissues is bound to the thylakoid membrane, it remains inactive. However, when the plant tissue undergoes damage, PPO becomes activated, leading to the oxidation of phenolic compounds with oxygen, resulting in the formation of quinones and subsequent browning of the explants [36]. The accumulation of quinones further intensifies the toxicity to the explants, ultimately leading to explant death [37]. In this study, the appropriate concentration of SNP effectively inhibited PPO activity, thus reducing the degree of browning. Phenolic acids and flavonoids are crucial components of phenolic compounds. They not only contribute to callus formation but also help maintain ROS balance through their potent antioxidant and free radical scavenging abilities [38,39]. Additionally, previous research has shown that SNP treatment can inhibit the surface browning of fresh-cut apples by suppressing the synthesis of phenols, thereby reducing the total phenol content and promoting the accumulation of flavonoids [40]. In this study, we observed a rapid increase in flavonoid content in the stem tip explants of *Malus sieversii* following SNP treatment, which aligns with the overall change in the total phenol content. These findings suggest that flavonoids are key factors influencing the browning of *Malus sieversii* stem tips.

## 4. Materials and Methods

### 4.1. Test Materials and Their Treatment

In this study, stem tip explants were derived from 1-year-old *Malus sieversii* seedlings for the development of an in vitro micropropagation system. The explants were prepared in accordance with Tan and Liu's sterilization protocol to ensure aseptic conditions [41]. The procedure commenced with a 30 min rinse under tap water, succeeded by surface sterilization using 70% ethanol for 30 s and a 5 min immersion in 1‰ mercuric chloride ($HgCl_2$). The explants were then washed five times with sterile distilled water [42]. The growth medium employed was the Murashige and Skoog (MS) formulation, modified by adjusting the plant hormone levels following Zhang's prescribed method [43]. This revised medium was enriched with 0.5 mg/L 6-BA and 0.5 mg/L NAA. Different concentrations of SNP solutions (0, 10, 50, 100 μM) were prepared. After the medium temperature reduced to 60 °C, sterilized and filtered SNP was added to the high-pressure sterilized medium (121 °C, 20 min). The medium was used immediately after it solidified. The culture conditions were as follows: light intensity of 5000 lx, a temperature of 25 °C during the day for 14 h, a temperature of 23 °C during the night for 10 h, and a relative humidity of 75%. Each tissue culture bottle contained three explants, and each treatment consisted of three tissue culture bottles. All procedures were repeated three times.

### 4.2. Determination of Browning Index

The extent of browning in the stem tip explants was assessed using a browning index, which was categorized into five grades. The classification of browning was as follows: grade 0 represented a white medium and a green stem tip incision, grade 1 represented a yellowish medium and a light brown stem tip cut, grade 2 represented a yellow medium and a reddish-brown stem tip cut, grade 3 represented a dark yellow medium and a dark brown stem tip incision, and grade 4 indicated complete browning of all buds and subsequent death of the materials. The browning index (BD) was calculated using the formula: BD = ($\sum$ (browning stage $\times$ number of plants)/total number of plants) $\times$ maximum score $\times$ 100%.

### 4.3. Determination of the Activity of Antioxidant Enzymes

A fresh sample weighing 0.5 g was taken. Then, 1 mL of phosphate buffer (0.05 M, pH 7.8) was added to the sample. After that, the sample was ground in an ice bath, and an additional 1 mL of buffer was added. The resulting mixture was transferred to a centrifuge tube. The mortar was washed with 2 mL of buffer, and this solution was poured into the centrifuge tube. The tube was balanced and then centrifuged for 20 min at a low temperature of 0–4 °C, with a speed of 10,500 rpm. Finally, the supernatant was collected for the determination of enzyme activity.

The activity of superoxide dismutase (SOD) was assessed using the nitrogen blue tetrazolium (NBT) colorimetry method described by Tajner [44]. One enzyme activity unit was defined as the amount of enzyme required to inhibit 50% of the photochemical reduction of NBT. The activities of peroxidase (POD) and catalase (CAT) were measured using the method outlined by Dong [45]. The guaiacol colorimetric method was used to determine POD activity, with the OD (optical density) value at 470 nm served as a unit of enzyme activity (U). CAT activity was determined using the ammonium molybdate colorimetry method, with the OD value at 240 nm used for measurement. The activity of ascorbate peroxidase (APX) was determined based on the method developed by Roach [46], with the OD value at 290 nm being recorded.

### 4.4. Determination of Non-Enzymatic Antioxidants

Proline (Pro) content was measured using the sulfosalicylic acid colorimetric method, with the OD value at 520 nm being recorded. This method was based on the procedure described by Shabnam [47]. The content of soluble sugar (SS) was determined using the plant soluble sugar kit from Suzhou Keming Biotechnology Co., Ltd.,Suzhou, China. The OD value at 620 nm was measured according to the instructions provided by the manufacturer. Soluble protein (SP) was determined using the Coomassie brilliant blue Gmur250 staining method, following the protocol outlined by Hang [48]. The OD value at 595 nm was determined using a bovine serum albumin standard curve.

### 4.5. Determination of Markers of Oxidative Stress

The content of malondialdehyde (MDA) was determined using the thiobarbituric acid colorimetry method, following the procedure described by Gong [49]. The contents of $H_2O_2$ and $O_2^-$ were measured using the biochemical kit from Suzhou Keming Biotechnology Co., Ltd.,Suzhou, China. OD values were recorded at 415 nm and 570 nm, respectively. The activity of PAL and PPO was determined using related kits from Suzhou Keming Biotechnology Co., Ltd., Suzhou, China. The content of total phenols was determined based on the Folin–Ciocerteu program described by Zhu [50]. The determination of total flavonoids was performed following the protocol outlined by Budiawan [51]. Fresh samples were extracted with 80% acetone, and OD values were measured at 510 nm.

### 4.6. Statistical Analysis

All data were analyzed using SPSS 21.0 software (IBM, Armonk, New York, NY, USA). The mean values of each treatment were compared using Duncan's multiple comparison test. Graphs were created using Origin 8.0 software (OriginLab, Northampton, MA, USA).

## 5. Conclusions

In summary, explants generate ROS post-incision, and an excess of ROS can lead to irreversible oxidative damage. This damage may impair the antioxidant system, leading to browning. The findings of this study suggest that exogenous SNP plays a beneficial role in mitigating short-term oxidative stress. NO is effective in alleviating the toxicity associated with this stress; however, higher concentrations of NO do not enhance this protective effect and may indeed be toxic themselves. Overall, exogenous NO appears to act as a regulatory signal, modulating the antioxidant defense response in stem tip explants. Yet, the specific molecular mechanisms underpinning this regulation require further investigation. These insights contribute to a deeper understanding of the role of NO in the physiological responses of plants to stress.

**Author Contributions:** Conceptualization, C.Y. and J.L.; methodology, X.Q.; software, Y.L.; validation, M.S. and Y.Z.; formal analysis, Y.H.; investigation, X.Q.; resources, X.S.; data curation, J.L.; writing—original draft preparation, C.Y.; writing—review and editing, Y.M.; visualization, C.Y.; supervision, X.S.; project administration, X.S.; funding acquisition, X.S. All authors have read and agreed to the published version of the manuscript.

**Funding:** This research was funded by a project supported by National Natural Science Foundation of China, grant number 32072520; the Fruit innovation team project of Shandong Province (CN), grant number SDAIT-06-07; and the Natural Science Foundation of Shandong Province (CN), grant number ZR2020MC132.

**Data Availability Statement:** Data are contained within the article.

**Conflicts of Interest:** The authors declare no conflict of interest.

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
