# Peer review of "Effect of Nitric Oxide on Browning of Stem Tip Explants of Malus sieversii"

_horticulturae, doi:10.3390/horticulturae9111246_

Round 1

Reviewer 1 Report

Comments and Suggestions for Authors

The manuscript submitted by Yang and collaborators reports the effects of exogenous oxide on the browning of Malus siversii explants during in vitro culture. The research design and results are adequately well described; however, discussion is a weak point in the manuscript. Main issue found: According to the data, the 50 uM SNP was the best dose inducing a protective effect; however, it is an apparent dose-response effect that the authors are considering. The lowest 10 uM decreases the browning index and triggers the enzymatic and non-enzymatic antioxidant systems; the 50 uM was better than 10 uM, but 100 uM seems to be an excess. As discussed in the manuscript submitted, 50 uM is a magic dose.

Moreover, the document presentation has to be improved (superscript not used in the authors list, scientific names are not italicized, there are some typos, and different fonts are used in the References section). Consider increasing the figure sizes; some are too small (fig. 3 and 4).

Comments on the Quality of English Language

Minor editing of English language required.

Reviewer 2 Report

Comments and Suggestions for Authors

The work aimed to investigate the impact of  exogenous NO donor Sodium nitroprusside (SNP) on the browning of Malus sieversii stem tip explants and the conclusions are proved by the conducted experiment.

Generally, the experiment is clear described, however the graphically must be improved.

The methods must be described substantially. The references are not sufficiently because of the specific of work and in vitro procedures. 

The conclusion should be rethinking - this part shouldn't not repeat results.

Patents - I think this part is not required in this research?

Comments on the Quality of English Language

The language / text should be more flow.

Round 2

Reviewer 1 Report

Comments and Suggestions for Authors

Dear authors,

At this point, I have no more issues about the revised manuscript. Only a minor point: 

- Remember, scientific plant names have be written in italics. Malus sieversii is in italics only in the main title of this new version of your manuscript.

Reviewer 2 Report

Comments and Suggestions for Authors

Thank you.

Comments on the Quality of English Language

It's OK.